# Birotons and "Dark" Quantum Hall Hierarchies

Oleg A. Grigorev [1], Liliya I. Musina [1,*], Alexander B. Van'kov [2], Oleg V. Volkov [1] and Leonid V. Kulik [1]

[1] Institute of Solid-State Physics Russian Academy of Sciences, 142432 Chernogolovka, Russia
[2] Laboratory for Condensed Matter Physics, HSE University, 101000 Moscow, Russia
* Correspondence: musina.li@phystech.edu

**Abstract:** A computational scheme is suggested to estimate neutral excitation energies in the fractional quantum Hall effect (FQHE) states. The FQHE states are systematized according to the Farey-number hierarchy structure. We show that besides the widely known Laughlin–Jain hierarchy of fractional states, there exist other "dark" hierarchies. Although hardly observed in the highest mobility samples, they can significantly affect the thermodynamics and spectral characteristics of the FQHE states. The known problems in the interpretation of the FQHE's experimental results are explained in terms of the coexistence of two fundamentally different transformations of the electron system, one of which is a neutral excitation in the FQHE state, whereas the other is a transition between two FQHE ground states, one of which represents the Laughlin–Jain FQHE hierarchy and the other a state of "dark" hierarchies.

**Keywords:** fractional quantum Hall effect; exact diagonalization; Farey series





## 1. Introduction

Today, the FQHE provides the only experimentally accessible system for observing anyons—quasiparticles with non-Fermi and non-Bose statistics. The pioneering works on the experimental detection of quasiparticles with abelian anyonic statistics $\pi/3$ in the FQHE state of 1/3 [1,2] opened fundamentally new prospects for incorporating anyons into applied physics. At present, the FQHE states with non-abelian anyons in the fractional states of 5/2 and 12/5 have been the focus of intense research [3,4]. In addition to single anyons, multiparticle anyon complexes have been observed, and their collective properties have been investigated [5]. In the very near future, further advancements in the experimental methods of studying anyon matter may well enable the observation of quasiparticles with more sophisticated abelian and non-abelian statistics than that of $\pi/3$. However, at the current stage of development, FQHE physics is facing challenges related to the thermodynamics and spectral properties of the observed fractional states. These issues raise fundamental questions about the FQHE hierarchical structure and the interrelation between different fractional states. For example, in the spectrum of the neutral excitations of the 1/3, 2/5, and 3/7 fractional states, there have been experimentally observed charge density excitations with abnormally low energies compared to the calculations [6]. Other significant problems arise concerning the interpretation of conductivity activation dependencies for the known FQHE states, close to the filling factor of 1/2 [7].

Creating a general hierarchical structure of fractional states requires building its foundation on the lowest spin sublevel of the zero Landau level $\nu < 1$. However, such a task can be considered within several theoretical approaches. The choice of a particular approach shall be based on either experimental observations or the numerical solution of the Schrodinger equation obtained by exact diagonalization for the system of a sufficiently large number of particles. In the present work, we argue that even in the best existing samples, the random potential precludes the observation of the vast majority of possible fractional states. For this reason, the exact diagonalization technique remains the only feasible method of organizing FQHE states into a hierarchical system.

All modern conceptions of FQHE states at the lowest Landau level can be summarized by the following well-established facts. First, there exist fractional states belonging to the main Laughlin hierarchy $\nu = \frac{1}{m}$ (where $m$ is an odd integer number), as well as their symmetrical counterparts, $\nu = 1 - \frac{1}{m}$, emerging due to the electron–hole symmetry. Second, there exists the Jain series. It generalizes the main Laughlin series with the electron filling factors expressed as $\nu = \frac{m}{2nm \pm 1}$ and their symmetric pairs of the form $1 - \frac{m}{2nm \pm 1}$ (where $n$ is an integer number). Finally, several experimentally detectable "weak" fractional states lie outside the main Laughlin–Jain hierarchy, for example, 4/11, 4/13, 5/13, etc.

Hence, it is natural to ask the question of whether the Laughlin–Jain hierarchy is a complete representation of fractional states, with few possible exceptions, or are there any fractional hierarchies for some reason unobservable experimentally ("dark hierarchies"). Indeed, since the very discovery of the FQHE, several theoretical approaches have been developed to answer this question. The works of Haldane and Halperin suggest the following model to account for possible hierarchies of the FQHE. When a large quantity of the charged excitations of the $\frac{1}{m}$ Laughlin state carrying the charge $\pm \frac{e}{m}$ (quasi-electrons and quasi-holes) are present in an electronic system, they themselves form a Laughlin-like child state at $\nu = \frac{p}{q}$ [8,9]. Since the quasiparticle charge reduces as the denominator increases proportionally to $q^{-1}$, the given Haldane–Halperin models predict a decrease in the value of the energy gap between the ground and the excited states with the growing denominator of the fractional state.

An alternative hierarchical structure of the FQHE states was proposed by Jain; he developed the conception of composite fermions—quasiparticles consisting of an electron and an even number of the magnetic flux quanta [10]. This model is based on mapping FQHE states onto composite fermions' integer QHE states (the Laughlin–Jain hierarchy mentioned above). The given structure can be generalized as the FQHE states with the filling factor $\nu$ are mapped onto the new fractional states with the electron filling factor $\nu_2 > 1$. Hence, successive mappings lead to one of the Laughlin states—the foundation of every hierarchy of the FQHE states. Such a construction corresponds to the system with several "sorts" of composite fermions that differ in the number of attached flux quanta. Generally speaking, given an arbitrary fractional filling factor $\nu$, both described theoretical approaches permit a few ways of bringing a particular FQHE state to the top of its hierarchy. Jain himself suggested selecting real FQHE states on the basis of energy consideration, whereas other studies argued that electron density configurations obtained in different ways are equivalent [11].

To resolve such an uncertainty, Zang and Birman [12] combined the approaches proposed by Halperin, Haldane, and Jain and devised a method of matching every fraction to a unique path leading to the top of its hierarchy. Not only does it remove the construction ambiguity, but it also enables qualitative estimation of the energy of the first excited state for every filling factor. In essence, the Zang–Birman arrangement is similar to Jain's, except that at every level of the hierarchy, a new composite particle "captures" two extra magnetic flux quanta. Thus, for the child state, the filling factor $\nu$ can be described by the mediant of the parent state, $\frac{p}{q}$, and the critical fraction, $\frac{p'}{q'}$, $q' = 2m$, as follows: $\nu = \frac{p}{q} \oplus \frac{p'}{q'} = \frac{p+p'}{q+q'}$. In this case, the energy gap that separates the ground state from the excited states drops for every descendant fraction within its own hierarchy. This rule does not impose principal restrictions on energy gaps in different hierarchies. Hence, there is no direct link between the denominator value of an arbitrary FQHE state and the energy gap. Finding the pattern in energy gaps for a number of filling factors can give us evidence on the credibility of a given hierarchy scheme, even without conjecturing a trial wavefunction for a state at said filling factors, as, e.g., Jain in [10]; therefore, it is important to find the gaps.

The choice between the given hierarchical models of the FQHE depends on the development of modern computational resources. However, even for the existing ultra-fast computers, calculating the dispersion dependencies of neutral excitations by solving the many-electron Schrodinger equation with exact diagonalization method proves difficult. It allows covering the range where the momentum corresponding to the energy gap lies

only for the main Laughlin states 1/3 and 2/3. Indeed, explicitly calculating the gap even for the next fractional state 2/5 in the Laughlin–Jain hierarchy presents a considerable computational challenge [13]. To overcome this technical difficulty, we used general assumptions about the dispersion dependencies of neutral excitations for the FQHE states at the zero Landau level regarding the link between the magnitude of the energy gap and the lowest energy at zero momentum. We tested these assumptions numerically for several fractions and built the hierarchy of the FQHE states for $\nu$ from 1/4 to 3/4—most suitable for the experimental study. This hierarchy was found to be in good agreement with the Zang–Birman model (Farey-number hierarchy structure [14]).

In Section 1, a computational technique is given—to obtain an energy spectrum of a many-electron system interacting via Coulomb repulsion in a cell with the periodic boundary condition, the exact diagonalization technique was used. The results of numerical experiments are listed in Section 2. In Sections 3 and 4, a possible explanation for the pattern that can be seen in the result of our numerical experiment is proposed and discussed.

## 2. Materials and Methods

We considered a two-dimensional system of electrons with Coulomb interaction confined to a parallelogram cell $\Lambda \ni z = \alpha\tau_1 + \beta\tau_2$, $0 \le \alpha, \beta \le 1$, $z$ being the complex coordinate and $\tau_{1,2}$ corresponding to coincident sides of the cell, and quantizing magnetic field $B$ perpendicular to its plane. We applied periodic boundary conditions (PBCs), that is the state of the system is conserved by magnetic translation by $\tau_{1,2}$. The PBCs are compatible if the cell is pierced by an integer number $N_s$ of magnetic flux quanta. In such a system, a Hamiltonian of a single electron has a well-known spectrum of $\hbar\omega_C\left(n + \frac{1}{2}\right)$, $\omega_C = \frac{eB}{m}$ (Landau levels), each energy level having a finite degeneracy equal to $N_s$. Therefore, for a number of electrons $N_e$, we can define the following base of states diagonalizing the kinetic part of Hamiltonian $\prod_{j=1}^{N_e} a^\dagger_{i_j,n_j}|0\rangle$, the pairwise interaction part given by

$$\hat{H}_C = \sum_{\alpha,\beta} \sum_{i_1,i_2,i_3,i_4} \sum_{n_1,n_2,n_3,n_4} V^{n_1 n_2 n_3 n_4}_{i_1 i_2 i_3 i_4} a^\dagger_{\alpha,i_1,n_1} a^\dagger_{\beta,i_2,n_2} a_{\beta,i_3,n_3} a_{\alpha,i_4,n_4} \tag{1}$$

where $a^\dagger_{\alpha,i_k,n_k}$ and $a_{\alpha,i_k,n_k}$ stand for, respectively, the creation and annihilation operators of a spin $\alpha$ electron $k$ in the state $\psi^n_i$, $n$ being the number of the Landau level and ranging from 0 to $\infty$ and $i$ specifying the number of the states within the Landau level and ranging from 1 to $N_s$.

To find the energy spectrum of the given system, it is necessary to calculate matrix elements of Coulomb potential and to diagonalize (1). We assumed that the cyclotron energy $\hbar\omega_C$ and Zeeman splitting are much larger than the Coulomb energy $\frac{e^2}{\epsilon l_B}$; hence, we can neglect the contribution of higher Landau levels and assume that all electrons are spin polarized; therefore, we only need a finite number of matrix elements for exact diagonalization (in what follows, we however present expressions for matrix elements for arbitrary pairs of Landau levels for the sake of completeness). For a multi-electron system with $N_e$ electrons occupying $N_s$ possible states, the allowed Fock basis comprises $C^{N_e}_{N_s}$ vectors, which makes the exact diagonalization for even simple fractions quite a tedious task. However, an observation due to Haldane [15] helps to build a simpler scheme, as there exists a certain momentum-like operator with $\gcd(N_e, N_s)$ quantum numbers.

The periodic eigenfunctions of Landau Hamiltonian were built by Yoshioka [16], who also applied the exact diagonalization scheme in the case of a rectangular cell, and later by Haldane and Rezayi [17], the matrix element for Coulomb interaction was computed in the case of a rectangular cell in [18]. However, the formula for a general parallelogram was never published, to our knowledge. Some results concerning the reciprocal vector operator were rederived in a fashion that seems more fitting for the problem.

### 2.1. Magnetic Translations and One-Electron Wavefunctions

For a single electron in a constant magnetic field, the Hamiltonian $\hat{H} = \frac{1}{2m}\left(\hat{p} - \frac{e}{c}A\right)^2$. We work in the Landau gauge $\hat{A} = \binom{0}{Bx}$, so one-electron states at the lowest Landau level can be represented as [16]

$$\psi(x,y) = f(z)\exp\left(-\frac{y^2}{2l_B^2}\right), \ z = x + iy \tag{2}$$

Because of the PBC, we have $\hat{t}_m(\tau_{1,2})\psi = \psi$, where magnetic translations:

$$\hat{t}_m(\tau) = \exp\left(\frac{i}{\hbar}\left(\hat{p} - \frac{e}{c}A\right)\tau\right). \tag{3}$$

By the Campbell–Hausdorff formula (here, $\hat{t}(\tau)$ stands for an ordinary translation by $\tau$ and $\tau = \tau_x + i\tau_y$):

$$\hat{t}_m(\tau) = \exp\left(\frac{i\tau_x\tau_y}{2l_B^2}\right)\exp\left(\frac{i\tau_y x}{l_B^2}\right)\hat{t}(\tau) \tag{4}$$

$$\hat{t}_m(\tau_1)\hat{t}_m(\tau_2) = \exp\left(\frac{i(\tau_{1x}\tau_{2y} - \tau_{1y}\tau_{2x})}{l_B^2}\right)\hat{t}_m(\tau_2)\hat{t}_m(\tau_1) \tag{5}$$

For a state to be conserved by two magnetic translations by $\tau_1$ and $\tau_2$, they must commute; therefore:

$$\tau_{1x}\tau_{2y} - \tau_{1y}\tau_{2x} = 2\pi N_s l_B^2, \ N_s \in \mathbb{Z} \tag{6}$$

Applying (4) to (2) and taking $\hat{t}_m(\tau_{1,2})\psi = \psi$ into account, we have the following condition on $f$:

$$\frac{f(z+\tau_i)}{f(z)} = \exp\left(-\frac{i(\tau_i + 2z)\tau_{iy}}{2l_B^2}\right) \tag{7}$$

Specifying $\tau_1 \in \mathbb{R}$, $\tau_2 = |\tau_2|e^{i\theta}$ (note that (6) yields $\tau_1|\tau_2|\sin\theta = 2\pi N_s l_B^2$), we obtain $\frac{f(z+\tau_1)}{f(z)} = 1$ and, hence, represent $f$ as a Fourier series $f(z) = \sum_{k\in\mathbb{Z}} c_k \exp\left(i\frac{2\pi kz}{\tau_1}\right)$. To find its coefficients, substitute it into (7), which gives

$$c_{k+N_s} = \exp\left(\frac{i\tau_2\pi N_s}{\tau_1}\right)\exp\left(i\frac{2\pi k\tau_2}{\tau_1}\right)c_k \tag{8}$$

Consequently $f(z)$ corresponding to LLL states with PBCs constitutes a linear space of dimension $N_s$, with one example of a base delivered by

$$f_r(z) = \sum_{m\in\mathbb{Z}} \exp\left(i\frac{m(m|\tau_2|\sin\theta + 2X_r)|\tau_2|e^{i\theta}}{2l_B^2}\right)\exp\left(i\frac{(X_r + m|\tau_2|\sin\theta)z}{l_B^2}\right), \tag{9}$$

$X_r = \frac{2\pi l_B^2 r}{\tau_1}$.

Finally, the LLL wavefunctions are

$$\psi_r(x,y) = \left(\frac{1}{\tau_1\sqrt{\pi}l_B}\right)^{\frac{1}{2}} \sum_{m\in\mathbb{Z}} \exp\left(i\frac{(m^2|\tau_2|^2\sin\theta + 2X_r m|\tau_2|)\cos\theta}{2l_B^2}\right) \times$$
$$\times \exp\left(i\frac{(X_r + m|\tau_2|\sin\theta)x}{l_B^2}\right)\exp\left(-\frac{(y + X_r + m|\tau_2|\sin\theta)^2}{2l_B^2}\right). \tag{10}$$

Similarly, at the *n*-th LL ($H_n$ is the *n*-th Hermite polynomial):

$$\psi_r^n = \left( \frac{1}{L_1 \sqrt{\pi} 2^n n! l_B} \right)^{\frac{1}{2}} \sum_{m \in \mathbb{Z}} e^{im\left(2\pi j \frac{L_2 \cos\theta}{L_1}\right)} e^{i\frac{m^2 L_2^2 \sin 2\theta}{4l_B^2}} \exp\left( i \frac{(X_r + mL_2 \sin\theta)x}{l_B^2} \right)$$

$$\exp\left( -\frac{(y + X_r + mL_2 \sin\theta)^2}{2l_B^2} \right) H_n\left( \frac{y + X_r + mL_2 \sin\theta}{l_B} \right) \quad (11)$$

### 2.2. Reciprocal Vectors and Partial Translations

For an operator to constitute an observable, it should commute both with the Landau Hamiltonian and magnetic translations $\hat{t}_m(\tau_{1,2})$. A remarkable observation due to Haldane [15] was that there exists center mass magnetic translation operators $\hat{T}(a) = \prod_{i=1}^{N_e} \hat{t}_{m,i}(a)$, where $\hat{t}_{m,i}(a)$ acts on the $i$-th electron, to nontrivial vectors $a$ satisfying both of these properties and also commuting with each other, thus constituting a vector quantum number similar to momentum.

Indeed, it follows from (4) that $\hat{T}(a)\hat{T}(b) = \exp\left( \frac{iN_e(a_x b_y - b_x a_y)}{l_B^2} \right) \hat{T}(b)\hat{T}(a)$. Consider $N = \gcd(N_e, N_s)$, then for $Na = \tau_1$, $Nb = \tau_2$ $[\hat{T}(a), \hat{T}(b)] = 0$:

$$\hat{T}\left(\frac{\tau_1}{N}\right) \hat{T}\left(\frac{\tau_2}{N}\right) = \exp\left( \frac{i2\pi N_e N_s}{N^2} \right) \hat{T}\left(\frac{\tau_2}{N}\right) \hat{T}\left(\frac{\tau_1}{N}\right)$$

What is the spectrum of these operators? Because each $\prod_{i=1}^{N_e} a_{j_i}^\dagger |0\rangle$ is an eigenstate for $\hat{T}\left(\frac{\tau_1}{N}\right)^N$, the $\hat{T}\left(\frac{\tau_1}{N}\right)$ eigenvalues are $\exp\left(\frac{2\pi i l}{N}\right)$. Two translation operators commute; hence, they have a common eigenstate base, the eigenvalues being $\exp\left(\frac{2\pi i l_i}{N}\right)$ for $\hat{T}(L_i)$. Consider $q$ such that $(q, L_1) = 2\pi l_1$, $(q, L_2) = 2\pi l_2$ (i.e., $q$ belongs to a reciprocal lattice), then the spectrum of $\hat{T}(L)$ is $\exp(\frac{i(q,L)}{N})$. This $q$ is a reciprocal vector quantum number.

Direct calculation shows that the one-electron states deduced in the previous section satisfy $\hat{T}\left(\frac{\tau_1}{N}\right)\psi_r = \exp\left(\frac{2\pi i r}{N}\right)\psi_r$, $\hat{T}\left(\frac{\tau_2}{N}\right)\psi_r = \exp\left(i\frac{(2r+1)}{N} \cdot \pi \frac{|\tau_2| \cos\theta}{\tau_1}\right)\psi_{r + \frac{N_s}{N}}$. Using this, we constructed a base out of reciprocal momentum eigenstates, and to diagonalize (1), it is only needed to handle blocks corresponding to each eigenvalue.

We will now focus on the allowed values of $q$ because these will be the eigenvalues for the blocks, and their absolute values correspond to the points in the dispersion curve. As we mentioned, they lie in the reciprocal lattice $\{n_1\tau_1 + n_2\tau_2\}^{-1}$ and are defined up to a translation by an element of $\left\{ \frac{n_1\tau_1}{N} + \frac{n_2\tau_2}{N} \right\}^{-1}$. Therefore, the number of allowed values $(N^2)$ of the reciprocal number at some fixed filling factor can be increased by increasing $\gcd(N_e, N_s)$, which makes the computational complexity of our problem dramatically increase. Keeping $N_s$ fixed, we can still increase the number of different absolute values of allowed $q$ by considering a non-square cell ("lifting degeneracy" of two points, which are reflections in a diagonal) and the maximum absolute value of $q$ by choosing a non-rectangular cell (maximizing the diagonal of a parallelogram while keeping its area). The latter may be crucial in constructing roton minima for some fractions.

### 2.3. Matrix Element

Finally, we need to evaluate the matrix elements in (1). Using the translational invariance of wavefunctions, it can be rewritten as follows:

$$A_{j_1 j_2 j_3 j_4}^{n_1 n_2 n_3 n_4} = \int_\Lambda dr_1 \int_\Lambda dr_2 \psi_{j_1}^{n_1 *}(r_1)\psi_{j_2}^{n_2 *}(r_2)\tilde{V}(r_1 - r_2)\psi_{j_3}^{n_3}(r_2)\psi_{j_4}^{n_4}(r_1) \quad (12)$$

where $\tilde{V}$ stands for periodic continuation of $V$, $\tilde{V}(r) = \sum_{k_1, k_2} \frac{e^2}{|r + k_1\tau_1 + k_2\tau_2|}$. Being a double-periodic function, it can be also represented by the following Fourier series

$\tilde{V}(z) = \frac{2\pi l_B^2}{\sigma} \sum_{q \in L^{-1}} \frac{e^2}{q} \exp(i(q,r)))$, where $\sigma$ denotes the area of a primitive cell of the lattice and $L = \{k_1 \tau_1 + k_2 \tau_2, \ k_1, k_2 \in \mathbb{Z}\}$, and the series is summed over the reciprocal lattice $L^{-1}$, $\forall q \in L^{-1}$, $r \in L$ $(q,r) = 2\pi N$, $N \in \mathbb{Z}$.

To account for Coulomb weakening, we modified the Fourier components by introducing the geometrical form-factor $F(q)$, calculated using the profile of the envelope wave function of electrons in the lowest-dimensional quantization sub-band of the conduction band in GaAs and obtain the following more realistic expression:

$$\tilde{V}(z) = \frac{2\pi l_B^2}{\sigma} \sum_{q \in L^{-1}} \frac{e^2 F(q)}{q} \exp(i(q,r))), \tag{13}$$

Plugging in the expressions from (11) and (13) and with some math, we arrive at:

$$A_{j_1 j_2 j_3 j_4}^{n_1 n_2 n_3 n_4} = \frac{2e^2}{\sigma} \sum_{q \in L^{-1}} \frac{F(q)}{q} \mathcal{G}^{n_1, n_4}(q) \mathcal{G}^{n_2 n_3}(-q), \tag{14}$$

where ($L_n^k$ stands for the generalized Laguerre polynomial)

$$\mathcal{G}^{n_s n_t}(q) = \sqrt{\pi} \exp\left( i \frac{2\pi^2 \cot\theta l_B^2}{\tau_1^2} \left( j_s - j_t - \frac{q_x \tau_1}{2\pi} \right) \left( j_s + j_t - \frac{q_x \tau_1}{2\pi} \right) \right) \exp\left( i q_y \left( X_{j_s} + \frac{q_x l_B^2}{2} \right) \right) e^{-\left(\frac{1}{2} l q\right)^2} \times$$

$$\times \sqrt{\frac{\min(n_s, n_t)!}{\max(n_s, n_t)!}} \left( \frac{l(\text{sign}(n_t - n_s) q_x + i q_y)}{\sqrt{2}} \right)^{|n_s - n_t|} L_{\min(n_s, n_t)}^{|n_s - n_t|} \left( \frac{q^2 l^2}{2} \right) \delta'_{\frac{q_x \tau_1}{2\pi} + j_t - j_s} \tag{15}$$

The function $F(q)$ was calculated numerically for the actual parameters of the experimental sample.

## 3. Results

In our study, we considered a hypothetical two-dimensional electron system with physical parameters of GaAs/AlGaAs quantum wells to provide a direct comparison with experimental data. The calculations were carried out for a system of several electrons in toric geometry, using the electron wavefunctions introduced in [17], invariant with respect to magnetic translations. To check the correctness of the numerical results, we calculated the dispersion dependencies of the five lowest-energy neutral excitations for the main fractional states of the Laughlin–Jain hierarchy, 1/3 and 2/3, related by the electron–hole symmetry (Figure 1). The spectrum of neutral excitations consists of a magneto-roton branch (MR) [19,20] and a continuum of multi-roton states (MMR). At zero momentum, the roton branch merges with the continuum and cannot be detected as a distinct excitation branch. The principal result of these calculations is that at zero momentum, the lowest-energy excitation is a biroton (magneto-graviton (BMR(0)) [21]) with twice the energy of the roton minimum, which is consistent with the previously reported findings [22]. The application range of our computational scheme was specified based on the condition of building the roton minimum with the smallest number of electrons (Figure 1).

The next step in the numerical calculations involved constructing the dispersion dependency of five neutral excitations with the lowest energy for the second representative of the Laughlin–Jain hierarchy, the fractional QHE state of 2/5. Once again, it can be claimed, though with less certainty than in the case of the fractional state of 1/3, that at zero momentum, the lowest energy of neutral excitations corresponds to the doubled energy of the absolute minimum of the roton branch [23]. Numerical calculations for the other fractional states lead to the conclusion that at $\nu < 1$, the spectrum of the lowest-energy neutral excitations of any FQHE state is of the same type. It consists of a multi-roton continuum and a roton branch (with the number of minima depending on the fractional state) damped at zero momentum.

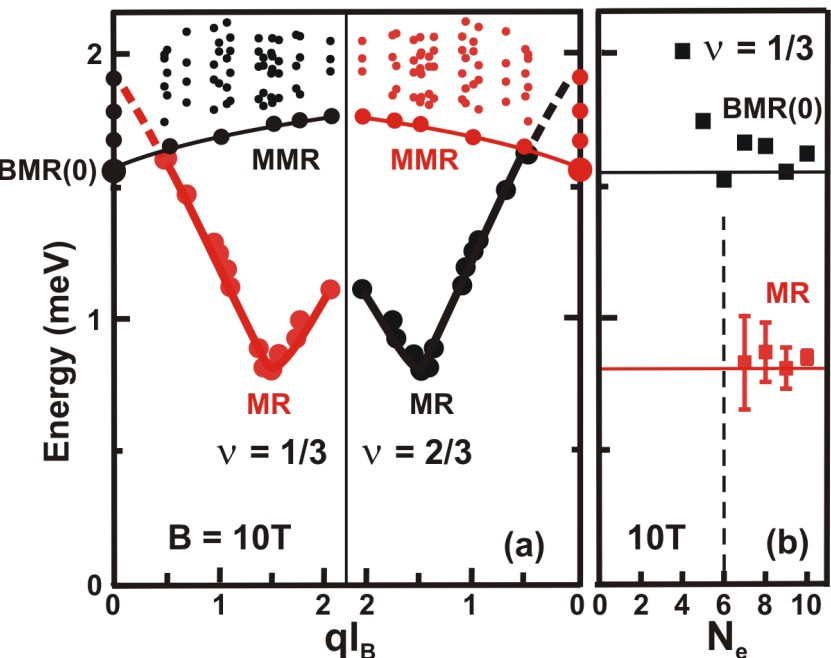

**Figure 1.** (**a**) Dispersion dependencies of the five lowest-energy excitations in Laughlin fractional QHE states 1/3 and 2/3, calculated for nine electrons in a $\delta$-function GaAs/AlGaAs quantum well at a magnetic field of 10 T. (**b**) Calculated energies of birotons with zero momentum and rotons in Laughlin state 1/3, at a magnetic field of 10 T, plotted versus the number of electrons. The dashed line indicates the smallest number of electrons needed to observe the minimum in the dispersion of the roton brunch. The solid lines are included as a matter of convenience.

Based on the calculations, it is natural to assume that the lowest-energy excitation with zero momentum is going to be a biroton with twice the energy of the absolute minimum of the roton branch. It corresponds to the appearance of two rotons with minimal possible energies and opposite momenta. In our analysis, we could also consider the question of the binding energy between the rotons in a biroton [13]. However, even for the main FQHE states 1/3 and 2/3, such energy turns out to be smaller than the numerical error of our simulations. Therefore, since calculating only the lowest energy of neutral excitations at zero momentum is far less complicated than finding the absolute minimum of a roton branch in the corresponding dispersion dependencies, it opens broad prospects for the comparative analysis of the excitation energies of different fractional states.

## 4. Discussion

Calculation results for biroton energies were compared to the activation dependencies of conductivity in fractional states of the Laughlin–Jain hierarchy. In this case, the doubled activation energy was considered approximately equal to the energy of a biroton with zero momentum [24]. Hence, we found that the activation energies closely agree with the calculation for the main fractions, 1/3 and 2/3. However, approaching the filling factor of 1/2 leads to increasing deviation between the calculated energies and the experimental values of activation energies. Thus, for example, for the fractional states of 5/11 and 6/11, the discrepancy between the calculated and experimental data already exceeds an order of magnitude (Figure 2). The observed disagreement cannot be explained by the non-locality of electronic wavefunctions due to the finite width of the quantum wells used in the experiment. Indeed, the change in biroton energies due to the quantum well width is virtually independent of the FQHE state (Figure 2). It should also be noted that in the experiment, a linear approximation of the activation energy of fractional states in the Laughlin–Jain hierarchy to the filling factor 1/2 leads to non-physical, negative values [7],

which does not occur in the numerical modeling (Figure 3). A plausible explanation of this effect is that, for electron filling factors in the vicinity of $\nu = 1/2$, we are observing a temperature-induced reorganization of electron density rather than activation dependency. In that case, the ground state is brought to the FQHE states of a close filling factor that belong to a "dark" (experimentally undetectable) FQHE state. The transition between different fractional states should be accompanied by the appearance of bulk conductivity in a two-dimensional electronic system. This effect is insignificant for the activation energies of the main fractional states of the Laughlin–Jain hierarchy, 1/3 and 2/3. Indeed, energy gaps between the ground states of these fractions and the "dark" fractional states of close filling factors is approximately equal to the activation energies themselves. However, as the electron filling factor approaches $\nu = 1/2$, the given energy gaps separating fractions of the Laughlin–Jain hierarchy and close to them "dark" fractional states are substantially reduced, leading to the decrease in "activation" energy observed in the experiment (Figure 2).

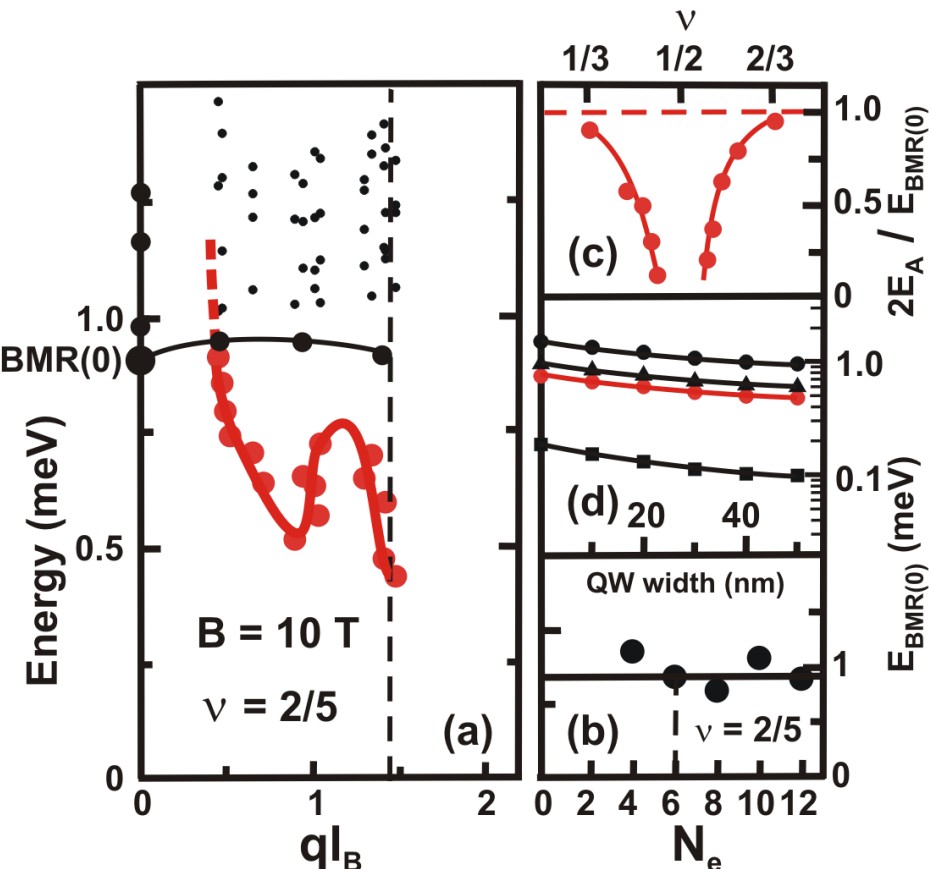

**Figure 2.** (**a**) Dispersion dependencies of the five lowest-energy excitations in the Laughlin fractional QHE state 2/5, calculated for twelve electrons in a $\delta$-function GaAs/AlGaAs quantum well at a magnetic field of 10 T. The dashed line marks the boundary of an elementary cell in inverse space. (**b**) Calculated energies of birotons with zero momentum for FQHE state 2/5 at a magnetic field of 10 T plotted as a function of the number of electrons. (**c**) The doubled experimental activation energy of fractional states from the Laughlin–Jain hierarchy measured in [7] (red dots) normalized by the calculated energies of birotons with zero momentum. (**d**) Energy dependencies of birotons with zero momentum for the FQHE states of 1/3 (black circles), 2/5 (black triangles), and 13/27 (black squares). For comparison, red circles indicate the energy dependency of the roton minimum for FQHE 1/3. The solid lines are included for convenience.

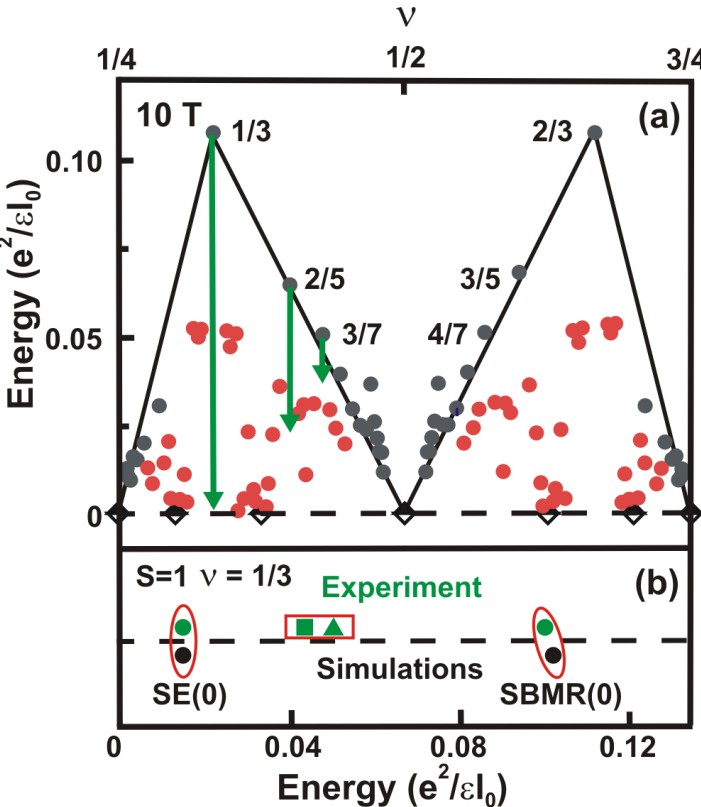

**Figure 3.** (**a**) Energies of birotons with zero momentum, at a magnetic field of 10 T, calculated for different FQHE states in the range of filling factors [1/4; 3/4], given a $\delta$-function GaAs/AlGaAs quantum well. Fractional states of the Laughlin–Jain hierarchy (black circles) and "dark" hierarchies (red circles) are expressed in Coulomb energy units. The solid lines are included for clarity. The length of green arrows signifies the measured charge density excitation energies from [6], corrected for the finite quantum well width and magnetic field used in the experiment. The diamonds denote the even electron filling factors, 1/4, 3/10, 3/8, 1/2, 5/8, 7/10 (left to right). (**b**) Energies of neutral excitations with spin 1 (S = 1) for the Laughlin state 1/3 calculated for spin birotons and spin excitons with zero momentum, at a magnetic field of 10 T (black dots). The green dots show the measured excitation energies from [5,6,25]. Red ovals indicate the corresponding pairs of experimental and calculated neutral excitations, SE(0)-spin exciton [6] and SBMR(0)-spin biroton (magneto-graviton) [5]. Squares mark the experimentally observable excitations that have no theoretical counterpart.

Another problem of FQHE states is the spectrum of experimentally detectable neutral excitations. Direct measurement of neutral excitations of the Laughlin–Jain hierarchy using resonant Raman scattering reproduces the pattern observed in the activation experiments [6]. The energy of neutral excitations for the FQHE state 1/3 shows excellent agreement with data calculated for a biroton (magneto-graviton) with zero momentum. In contrast, for the following fractional states of the Laughlin–Jain hierarchy, 2/5 and 3/7, the energies of the excitations decrease drastically, showing no resemblance to the calculation results. One possible way of explaining the observed effect is to take into account "dark" FQHE states, namely the optical transitions between the ground states of different FQHE states, as the local redistribution of the electron density induced by the electromagnetic field of a light wave. The coexistence of two different optical transitions: (i) between different fractional states and (ii) within a single fractional state were already discussed in [25].

The authors of the paper observed analogous abnormal optical transitions in the spectrum of spin density neutral excitations for the main Laughlin state 1/3, measured by the resonant reflection for several heterostructures with GaAs/AlGaAs quantum wells [5,26,27]. The spectrum indicates the direct optical transitions that excite spin birotons (spin magneto-gravitons), as well as the transitions that give rise to excitations with the energy lying in

the bandgap of the fractional Hall dielectric $\nu = 1/3$ (Figure 3). Unlike birotons, which exhibit bosonic properties predicted for neutral excitations [5], the anomalous excitations do not show signs of Bose statistics [26,27], suggesting possible transitions between the ground states of different fractional states rather than neutral excitations in the fractional QHE state 1/3.

## 5. Conclusions

In the conclusion of the study, we constructed the hierarchy of computable fractional QHE states in the range of electron filling factors from 1/4 to 3/4 (Figure 4). The energies of these fractional states were found to be reasonably consistent with the Zang–Birman hierarchical structure—Farey-number hierarchy structure [12]. Although this particular structure has much in common with the theoretical models of Halperin, Haldane, and Jain, it has a significant distinctive feature. Fractional states of highly varied denominator values from different FQHE hierarchies can have nearly equal energy gaps between the ground and excited states. For instance, the energies of such dissimilar fractional states as 6/19, 6/17, 8/23, 8/25, 10/29, and 10/31 belonging to different "dark" hierarchies are found to be almost the same, as all of them belong to the second steps from the top of their hierarchical ladders—the Laughlin state 1/3 (Figure 4). Conversely, the energies of fractional QHE states with equal denominators and similar electron filling factors can differ substantially. For example, the respective biroton energies of fractional states 10/31 and 11/31, corresponding to the second and third steps from the top of their hierarchical ladders, come to be nearly two-orders of magnitude apart ($51.8 \cdot 10^{-3}$ and $0.74 \cdot 10^{-3}$ in Coulomb units (Figure 3), respectively).

Despite the abundance of calculated fractional QHE states from "dark" hierarchies, the fact that they are not directly observable in magneto-transport experiments is quite understandable. The hierarchical structure proposed in the present work makes it evident that biroton energies of fractional states in the Laughlin–Jain hierarchy are always greater than other fractional states of comparable filling factors belonging to "dark" hierarchies. Hence, it is more energy favorable to localize some part of excited quasi-electrons and quasi-holes of Laughlin–Jain fractional states and to keep the filling factor of extended states unchanged in the region of filling factors separating the neighboring fractional states of this hierarchy. Considering that even a small amount of impurities in a two-dimensional system causes localization of a macroscopic number of electron states, the filling factor regions where we can observe the fractional states of "dark" hierarchies are not large, even in the most highly mobile samples known to this day [28]. Usually, for $\nu < 1$, these regions fall within narrow ranges of electron filling factors, [1/3; 2/5] and [3/5; 2/3], where the Laughlin–Jain fractional states are far apart in terms of the filling factor. As the electron filling factor approaches 1/2, free of Laughlin–Jain fractional states, regions shrink in size because the "separation" (in terms of the electron filling factor) between the nearby fractional states of the Laughlin–Jain hierarchy decreases as the inverse square of the fraction's denominator. Therefore, while the Laughlin–Jain hierarchy is not the only hierarchy of FQHE states, it is still dominant over the "dark" hierarchies in magnetotransport experiments (Figure 4). The "dark" hierarchies become, in turn, essential in describing the excitation properties of FQHE states.

The "dark" hierarchies are nevertheless essential in describing the excitation properties of FQHE states. Taking into account transitions between the FQHE states of the Laughlin–Jain hierarchy and the states of the "dark" hierarchies, the non-physical negative values for activation energies close to the filling factor 1/2 [7] are explained. The abnormal excitation energies measured by resonant Raman scattering for the fractional states, 2/5 and 3/7 [6], as well as the spin excitation energies for the state 1/3 measured by the authors of this paper [5,26,27] are similarly explained,.

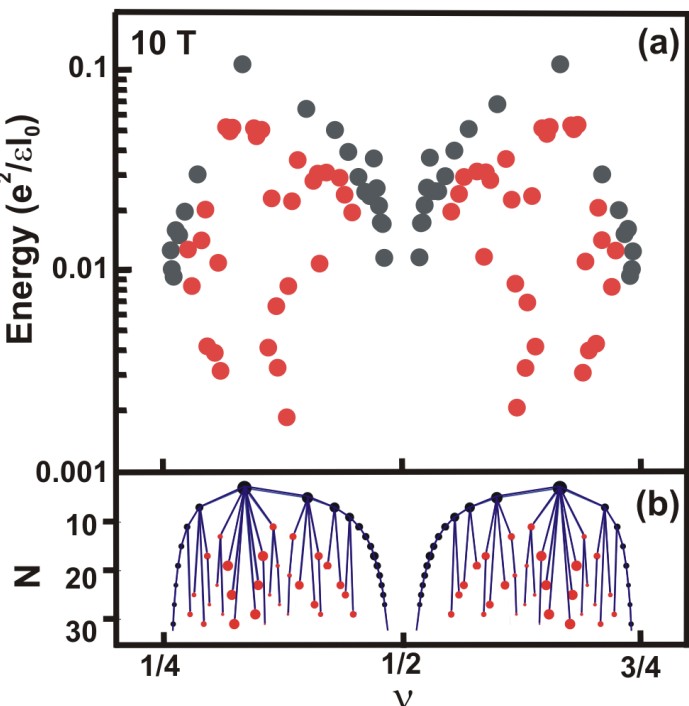

**Figure 4.** (**a**) Energies of birotons with zero momentum, in a magnetic field of 10 T, calculated for different FQHE states in the range of Coulomb energies from 0.001 to 0.1 $e^2/\epsilon l_0$. The fractional states of the Laughlin–Jain hierarchy and of the "dark" hierarchies are plotted in black and red circles, respectively. (**b**) FQHE states with energies from (**a**) brought into the Zang–Birman hierarchical structure (Farey-number hierarchy structure). Horizontal and vertical axes denote, accordingly, the fraction's absolute value and its denominator. The dot diameter is proportional to the energy of the biroton with zero momentum of the corresponding fraction. The lines link the fractional states of individual hierarchies.

**Author Contributions:** L.V.K. conceived of the experiments; L.I.M. and A.B.V. performed the numerical simulation; O.V.V. developed the program; O.A.G. developed the theoretical methods; O.A.G., L.I.M. and L.V.K. analyzed the results; L.V.K. and O.A.G. wrote the manuscript. All authors have read and agreed to the published version of the manuscript.

**Funding:** The work was supported by the Russian Science Foundation, Grant 18-12-00246.

**Data Availability Statement:** The processed numerical data files are available at the public GitHub repository (https://github.com/lilymusina/dark_hierarchies (accessed on 22 February 2022)).

**Conflicts of Interest:** The authors declare no conflict of interest.

## Abbreviations

The following abbreviations are used in this manuscript:

| | |
|---|---|
| QHE | Quantum Hall effect |
| FQHE | Fractional quantum Hall effect |
| PBC | Periodic boundary condition |

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
