# Peer review of "Birotons and “Dark” Quantum Hall Hierarchies"

_applsci, doi:10.3390/app12157940_

Round 1

Reviewer 1 Report

Referee's report on the MDPI article

Birotons and “dark” quantum Hall hierarchies

by Oleg A. Grigorev, Liliya I. Musina, Alexander B. Van’kov, Oleg V. Volkov and Leonid V. Kulik

The authors consider the hierarchical FQHE states, and argue that, by exploring the exact diagonalization technique for the lowest Landau level, one gets a series of “dark sequences” which are hardly observable in transport measurements but could be visible in optical and calorimetric experiments. The question itself, and the obtained results are potentially valuable and deserve publication. However, the presentation is in referee’s view not easy for reading and needs to be improved. The concrete remarks follow.

(p.2)

The sentences starting with “However, even for …” looks incomplete. Furthermore, saying at the end of introductory section “we used general assumptions …” the authors do not specify what concrete assumptions they have in mind. This part of the section does not contain the plan of the manuscript, but only statements which are more appropriate for a (very short) abstract. This is certainly inhibiting in the reading of the bulk of the text. Regarding the references, their numeration does not follow their chronological appearance in the text. Also, the citations are done in an unusual way, by skipping the page numbers.

(ps. 3, 4)

The authors introduce parallelogram (and not rectangular) unit cell, with a general angle Referee's report on the MDPI article

Birotons and “dark” quantum Hall hierarchies

by Oleg A. Grigorev, Liliya I. Musina, Alexander B. Van’kov, Oleg V. Volkov and Leonid V. Kulik

The authors consider the hierarchical FQHE states, and argue that, by exploring the exact diagonalization technique for the lowest Landau level, one gets a series of “dark sequences” which are hardly observable in transport measurements but could be visible in optical and calorimetric experiments. The question itself, and the obtained results are potentially valuable and deserve publication. However, the presentation is in referee’s view not easy for reading and needs to be improved. The concrete remarks follow.

(p.2)

The sentences starting with “However, even for …” looks incomplete. Furthermore, saying at the end of introductory section “we used general assumptions …” the authors do not specify what concrete assumptions they have in mind. This part of the section does not contain the plan of the manuscript, but only statements which are more appropriate for a (very short) abstract. This is certainly inhibiting in the reading of the bulk of the text. Regarding the references, their numeration does not follow their chronological appearance in the text. Also, the citations are done in an unusual way, by skipping the page numbers.

(ps. 3, 4)

The authors introduce parallelogram (and not rectangular) unit cell, with a general angle in expressions for the translations and for the corresponding wave functions and matrix elements (p. 4 and further on), pointing out that in their work for the first time this generalization has been done. However, it is not clear what was motivation for this step. From one side, it is not transparent how the variation of angle  does influence numerical results shown in Figs. 1 – 4. From the other side, since the authors compare their results (based on the „hypothetical two-dimensional electron system“ (p.6)) with experiments on GaAs/AlGaAs heterostructures, it remains unclear what is the connection of the crystal structure of these compounds and the geometry of the unit cell used in the present article.

Even more, it is as well unclear to what extent is the simple model used in the calculations and the analysis (electron band with circular symmetry and bare electron mass) relevant at all for the band electrons in these heterostructures, known to have e. g. much smaller, and anisotropic, band electron mass, etc.   

(p.3)

Looking at the eq.2, and equations that follow, one would guess that the authors use the Landau gauge for the vector potential. However, the referee was not able to find any information throughout the text whether, and how, the choice of gauge was made.  

(p. 6)

While the derivations of basis wave functions and accompanying matrix elements are rather detailed (Sec. 1, some critical remarks regarding this part are mentioned above), the details regarding the numerical part (Sec. 2) are mainly missing. Particularly absent are elements connected with Laughlin-Jain-Haldane-Yoshioka handlings in constructing FQHE states, announced in the introductory part. This inhibits the further reading of the working part of the article. This part, and the ensuing results are in fact effectively formulated in an informative manner and are mostly based on accompanying figures. Even the figures are not always simply readable; as an example, I was not able to recognize “five lowest-energy excitations” in Fig. 2a.

In conclusion, the article in its present form cannot be recommended for publication. Nevertheless, it could be acceptable after revisions which would meet the above critical remarks and would in general bring the work to the more transparent and complete level of presentationReferee's report on the MDPI article

Birotons and “dark” quantum Hall hierarchies

by Oleg A. Grigorev, Liliya I. Musina, Alexander B. Van’kov, Oleg V. Volkov and Leonid V. Kulik

The authors consider the hierarchical FQHE states, and argue that, by exploring the exact diagonalization technique for the lowest Landau level, one gets a series of “dark sequences” which are hardly observable in transport measurements but could be visible in optical and calorimetric experiments. The question itself, and the obtained results are potentially valuable and deserve publication. However, the presentation is in referee’s view not easy for reading and needs to be improved. The concrete remarks follow.

(p.2)

The sentences starting with “However, even for …” looks incomplete. Furthermore, saying at the end of introductory section “we used general assumptions …” the authors do not specify what concrete assumptions they have in mind. This part of the section does not contain the plan of the manuscript, but only statements which are more appropriate for a (very short) abstract. This is certainly inhibiting in the reading of the bulk of the text. Regarding the references, their numeration does not follow their chronological appearance in the text. Also, the citations are done in an unusual way, by skipping the page numbers.

(ps. 3, 4)

The authors introduce parallelogram (and not rectangular) unit cell, with a general angle theta in expressions for the translations and for the corresponding wave functions and matrix elements (p. 4 and further on), pointing out that in their work for the first time this generalization has been done. However, it is not clear what was motivation for this step. From one side, it is not transparent how the variation of angle theta   does influence numerical results shown in Figs. 1 – 4. From the other side, since the authors compare their results (based on the „hypothetical two-dimensional electron system“ (p.6)) with experiments on GaAs/AlGaAs heterostructures, it remains unclear what is the connection of the crystal structure of these compounds and the geometry of the unit cell used in the present article.

Even more, it is as well unclear to what extent is the simple model used in the calculations and the analysis (electron band with circular symmetry and bare electron mass) relevant at all for the band electrons in these heterostructures, known to have e. g. much smaller, and anisotropic, band electron mass, etc.   

(p.3)

Looking at the eq.2, and equations that follow, one would guess that the authors use the Landau gauge for the vector potential. However, the referee was not able to find any information throughout the text whether, and how, the choice of gauge was made.  

(p. 6)

While the derivations of basis wave functions and accompanying matrix elements are rather detailed (Sec. 1, some critical remarks regarding this part are mentioned above), the details regarding the numerical part (Sec. 2) are mainly missing. Particularly absent are elements connected with Laughlin-Jain-Haldane-Yoshioka handlings in constructing FQHE states, announced in the introductory part. This inhibits the further reading of the working part of the article. This part, and the ensuing results are in fact effectively formulated in an informative manner and are mostly based on accompanying figures. Even the figures are not always simply readable; as an example, I was not able to recognize “five lowest-energy excitations” in Fig. 2a.

In conclusion, the article in its present form cannot be recommended for publication. Nevertheless, it could be acceptable after revisions which would meet the above critical remarks and would in general bring the work to the more transparent and complete level of presentationin expressions for the translations and for the corresponding wave functions and matrix elements (p. 4 and further on), pointing out that in their work for the first time this generalization has been done. However, it is not clear what was motivation for this step. From one side, it is not transparent how the variation of angle  does influence numerical results shown in Figs. 1 – 4. From the other side, since the authors compare their results (based on the „hypothetical two-dimensional electron system“ (p.6)) with experiments on GaAs/AlGaAs heterostructures, it remains unclear what is the connection of the crystal structure of these compounds and the geometry of the unit cell used in the present article.

Even more, it is as well unclear to what extent is the simple model used in the calculations and the analysis (electron band with circular symmetry and bare electron mass) relevant at all for the band electrons in these heterostructures, known to have e. g. much smaller, and anisotropic, band electron mass, etc.   

(p.3)

Looking at the eq.2, and equations that follow, one would guess that the authors use the Landau gauge for the vector potential. However, the referee was not able to find any information throughout the text whether, and how, the choice of gauge was made.  

(p. 6)

While the derivations of basis wave functions and accompanying matrix elements are rather detailed (Sec. 1, some critical remarks regarding this part are mentioned above), the details regarding the numerical part (Sec. 2) are mainly missing. Particularly absent are elements connected with Laughlin-Jain-Haldane-Yoshioka handlings in constructing FQHE states, announced in the introductory part. This inhibits the further reading of the working part of the article. This part, and the ensuing results are in fact effectively formulated in an informative manner and are mostly based on accompanying figures. Even the figures are not always simply readable; as an example, I was not able to recognize “five lowest-energy excitations” in Fig. 2a.

In conclusion, the article in its present form cannot be recommended for publication. Nevertheless, it could be acceptable after revisions which would meet the above critical remarks and would in general bring the work to the more transparent and complete level of presentation

Author Response

(p.2)
The sentences starting with “However, even for …” looks incomplete. Furthermore, saying at the end of introductory section “we used general assumptions …” the authors do not specify what concrete assumptions they have in mind. This part of the section does not contain the plan of the manuscript, but only statements which are more appropriate for a (very short) abstract. This is certainly inhibiting in the reading of the bulk of the text. Regarding the references, their numeration does not follow their chronological appearance in the text. Also, the citations are done in an unusual way, by skipping the page numbers.

Response: Authors thank the Reviewer for a careful and attentive approach to the manuscript. We have changed the mentioned sentence, added the outline to the end of the introductory section and corrected the references.

(ps. 3, 4)
The authors introduce parallelogram (and not rectangular) unit cell, with a general angle theta in expressions for the translations and for the corresponding wave functions and matrix elements (p. 4 and further on), pointing out that in their work for the first time this generalization has been done.

Response: It was not stated in the manuscript that this is the first time a parallelogram cell was used, on the contrary, there were plethora of papers we referenced dating back to late 1980s where, say, a hexagonal cell was considered (Haldane 1990). However, the authors couldn't find a formula for the Coulomb potential matrix element on a arbitrary parallelogram cell, so it was decided to rederive it and include the explicit expression for it for the sake of integrity, reproducibility and potential further development of the research. 

(ps. 3, 4)
However, it is not clear what was motivation for this step. From one side, it is not transparent how the variation of angle theta   does influence numerical results shown in Figs. 1 – 4. 

Response: The Reviewer's concerns are clear, as, indeed, a big portion of the derivations is not used for the final results. However, choosing a more general cell should allow the researcher to lift the degeneracy when computing the dispersion curve and to access the more far away points without increasing the computational cost. There are plans to use these advantages in further work, as well as desire to share the calculations with scientific community. The clarifications on this matter are added to the text.

(ps. 3, 4)
From the other side, since the authors compare their results (based on the „hypothetical two-dimensional electron system“ (p.6)) with experiments on GaAs/AlGaAs heterostructures, it remains unclear what is the connection of the crystal structure of these compounds and the geometry of the unit cell used in the present article.

Response: By the term cell in the manuscript we mean computational cell, a model of an infinite 2D system restricted by boundary conditions. There is no connection to the crystal structure of material; moreover, the effects of a finite cell are artifacts that we try to get rid of. 

(ps. 3, 4)
Even more, it is as well unclear to what extent is the simple model used in the calculations and the analysis (electron band with circular symmetry and bare electron mass) relevant at all for the band electrons in these heterostructures, known to have e. g. much smaller, and anisotropic, band electron mass, etc.   

Response: We understand the concern of the referee, yet for our particular case of electrons in the conduction band of GaAs/AlGaAs quantum well all possible effects from non parabolicity and angle anisotropy are too small to take them into account. 

(p.3)
Looking at the eq.2, and equations that follow, one would guess that the authors use the Landau gauge for the vector potential. However, the referee was not able to find any information throughout the text whether, and how, the choice of gauge was made.  

Response: We pushed the explicit reference to the Landau gauge from line 130 back to where the one-electron wavefunction was introduced.

(p. 6)
While the derivations of basis wave functions and accompanying matrix elements are rather detailed (Sec. 1, some critical remarks regarding this part are mentioned above), the details regarding the numerical part (Sec. 2) are mainly missing. Particularly absent are elements connected with Laughlin-Jain-Haldane-Yoshioka handlings in constructing FQHE states, announced in the introductory part. This inhibits the further reading of the working part of the article. This part, and the ensuing results are in fact effectively formulated in an informative manner and are mostly based on accompanying figures. Even the figures are not always simply readable; as an example, I was not able to recognize “five lowest-energy excitations” in Fig. 2a.

Response:  There are no state wavefunctions explicitly constructed in this work. An extra clarification about our approach to the problem (from energy gaps point of view) is added to Introduction section. It is indeed a very interesting task, as it may be used e.g. for pair correlation function study; that is planned to be implemented in our upcoming works.
For anyone interested in raw data used for the figures a github repository is presented in the Data Availability Statement on page 12.
As for the Fig. 2a, the lowest of the five branches is distinct, while four others form a continuum and are in fact close to the point of mixing. 

Reviewer 2 Report

The manuscript titled by “Birotons and “dark” quantum Hall hierarchies”, reports an interesting theoretical calculation to estimate neutral excitation energies in the fractional quantum Hall effect (FQHE) states. The authors suggest that the “dark” hierarchies become in turn essential in describing the excitation properties of FQHE states.

The topic of the paper is quite interesting. However, some terms used in paper such as “Birotons” are not well clarified. These specific words make the paper hard to follow for general readers. The authors should improve the presentation of the paper so it will be clear for readers what exactly are the objectives and conclusions in this work.  What is the new physics in this work?

Author Response

Authors highly appreciate the Reviewer's comments on the manuscript. We believe the following corrections will make the work more clear and readable.

The explanation of the term  Biroton is included in the beginning of Results section. 

The new physics in our work is related to an opportunity to estimate and to compare quantitatively excitation gaps for fractional states of all possible fractional hierarchies and thereby to chose a right one amid existing hierarchy structures of fractional quantum Hall states. This idea is included in introductory section.

Round 2

Reviewer 1 Report

The authors considered the suggestions from my previous report and made accordingly changes in the manuscript. Although I still have the impression that the presentation and the organization of the material could be done in a more appropriate way (from the point of view of a reader), I understand  from their response that the authors consider this work (also) as a step towards further work on the problem of QHE hierarchies. In this respect the present text could have the purpose as a preliminary one, and as such is recommended for publication.

Reviewer 2 Report

I am satisfied with the authors' response. The paper can be published as its current form.